# Brazilian Multiethnic Association Study of Genetic Variant Interactions among *FOS, CASP8*, *MMP2* and *CRISPLD2* in the Risk of Nonsyndromic Cleft Lip with or without Cleft Palate

**DOI:** 10.3390/dj11010007

**Published:** 2022-12-26

**Authors:** Renato Assis Machado, Lilianny Querino Rocha de Oliveira, Ana Lúcia Carrinho Ayroza Rangel, Silvia Regina de Almeida Reis, Rafaela Scariot, Daniella Reis Barbosa Martelli, Hercílio Martelli-Júnior, Ricardo D. Coletta

**Affiliations:** 1Department of Oral Diagnosis, School of Dentistry, University of Campinas, Piracicaba 13414-018, São Paulo, Brazil; 2Hospital for Rehabilitation of Craniofacial Anomalies, University of São Paulo, Bauru 17012-900, São Paulo, Brazil; 3Graduate Program in Oral Biology, School of Dentistry, University of Campinas, Piracicaba 13414-018, São Paulo, Brazil; 4Center of Biological Sciences and of the Health, School of Dentistry, State University of Western Paraná, Cascavel 85819-110, Paraná, Brazil; 5Department of Basic Science, Bahiana School of Medicine and Public Health, Salvador 40290-000, Bahia, Brazil; 6Department of Oral and Maxillofacial Surgery, School of Health Science, Federal University of Paraná, Curitiba 80060-000, Parana, Brazil; 7Stomatology Clinic, Dental School, State University of Montes Claros, Montes Claros 39401-089, Minas Gerais, Brazil; 8Center for Rehabilitation of Craniofacial Anomalies, Dental School, University of Professor Edson Antônio Velano, Alfenas 37130-000, Minas Gerais, Brazil

**Keywords:** cleft lip, cleft lip and palate, single-nucleotide polymorphism, genetic epistasis

## Abstract

Associations of *CRISPLD2* (cysteine-rich secretory protein LCCL domain containing 2) and genes belonging to its activation pathway, including *FOS* (Fos proto-oncogene), *CASP8* (caspase 8) and *MMP2 (matrix metalloproteinase 2),* with nonsyndromic orofacial cleft risk, have been reported, but the results are yet unclear. The aim of this study was to evaluate single nucleotide polymorphisms (SNPs) in *FOS*, *CASP8* and *MMP2* and to determine their SNP-SNP interactions with *CRISPLD2* variants in the risk of nonsyndromic cleft lip with or without cleft palate (NSCL±P) in the Brazilian population. The SNPs rs1046117 (*FOS*), rs3769825 (*CASP8*) and rs243836 (*MMP2*) were genotyped using TaqMan allelic discrimination assays in a case-control sample containing 801 NSCL±P patients (233 nonsyndromic cleft lip only (NSCLO) and 568 nonsyndromic cleft lip and palate (NSCLP)) and 881 healthy controls via logistic regression analysis adjusted for the effects of sex and genomic ancestry proportions with a multiple comparison *p* value set at ≤0.01. SNP-SNP interactions with rs1546124, rs8061351, rs2326398 and rs4783099 in *CRISPLD2* were performed with the model-based multifactor dimensionality reduction test complemented with a 1000 permutation-based strategy. Although the association between *FOS* rs1046117 and risk of NSCL±P reached only nominal *p* values, NSCLO risk was significantly higher in carriers of the *FOS* rs1046117 C allele (OR: 1.28, 95% CI: 1.10–1.64, *p* = 0.004), TC heterozygous genotype (OR: 1.59, 95% CI: 1.16–2.18, *p* = 0.003), and in the dominant model (OR: 1.50, 95% CI: 1.10–2.02, *p* = 0.007). Individually, no significant associations between cleft risk and the SNPs in *CASP8* and *MMP2* were observed. SNP-SNP interactions involving *CRISPLD2* variants and rs1046117 (*FOS*), rs3769825 (*CASP8*) and rs243836 (*MMP2*) yielded several significant *p* values, mostly driven by *FOS* rs1046117 and *CASP8* rs3769825 in NSCL±P, *FOS* rs1046117 in NSCLO and *CRISPLD2* rs8061351 in NSCLP. Our study is the first in the Brazilian population to reveal the association of *FOS* rs1046117 with NSCLO risk, and to support that *CRISPLD2*, *CASP8*, *FOS* and *MMP2* interactions may be related to the pathogenesis of this common craniofacial malformation.

## 1. Introduction

Nonsyndromic cleft lip only (NSCLO) and nonsyndromic cleft lip and palate (NSCLP), that combined are denominated nonsyndromic cleft lip with or without cleft palate (NSCL±P), are the most common craniofacial defect in the world [1]. Around 1 out of 700 live births worldwide are affected, but large variations, depending on population origin, are observed [1]. Asians and Native Americans have the highest occurrence (1:500), followed by Europeans (1:1000), while Africans have the lowest incidence (1:2500) [2,3]. The incidence fluctuates between 1:650 and 1:2700 live births in Brazil due to the population’s ethnic diversity [4,5]. Despite the fact that NSCL±P has a multifactorial and complex genesis, our knowledge about the genetic and environmental players is still incomplete [6]. As a complex polygenic disorder with clear influence of the ancestral origin, it is essential to characterize the critical genetic variants in development pathways that interfere with normal lip and palate embryogenesis, defining common and population-specific risk variants and understanding the contribution for the etiology of malformation.

In an original genome-wide scan, Chiquet et al. revealed a strong linkage between *CRISPLD2* and multiplex families with NSCL±P [7]. Follow up studies confirmed that Crispld2 is expressed in the lateral *palatine* processes during palatogenesis, and single-nucleotide polymorphisms (SNPs) within the *CRISPLD2* gene are associated with NSCL±P risk [8,9]. In zebrafish, Crispld2 silencing impaired the neural crest cell migration, resulting in both jaw and palatal abnormalities in a dose-dependent manner [10], which was associated, among other putative candidates, with dysregulated expression of *FOS* (Fos proto-oncogene, which encodes a leucine zipper protein that can dimerize with proteins of the JUN family, thereby forming the transcription factor complex AP-1), *CASP8* (caspase 8) and *MMP2* (matrix metalloproteinase 2) [11]. Although putative roles for those genes are expected during orofacial development, only *MMP2* has been described as essential for normal palatogenesis [12]. In our previous study, associations of *CRISPLD2* rs4783099 and rs8061351 variants with NSCL±P were detected, with rs8061351 association being driven by participants with high African genomic ancestry [13]. Indeed, the influence of the intense admixed ancestry of the Brazilian population, mainly from three different roots (Amerindians, Europeans and Africans), in NSCL±P susceptibility has been previously reported [14,15,16].

In this study, we investigated the association of SNPs in *FOS* (rs1046117), *CASP8* (rs3769825) and *MMP2* (rs243836) with NSCL±P risk in a Brazilian population using an ancestry-structured case-control approach. The SNP-SNP epistatic interactions of *CRISPLD2* variants (rs1546124, rs8061351, rs2326398 and rs4783099), which were previously studied by us [13], with those SNPs on NSCL±P susceptibility were also assessed.

## 2. Materials and Methods

### 2.1. Ethics

The study was approved by the ethics review board of each of the centers affiliated with the collaborative study (approval number 08452819.0.0000.5418 on 29 April 2019). Written informed consent was obtained from the parents or guardians and/or the participants in compliance with the World Medical Association Declaration of Helsinki, Ethical Principles for Medical Research Involving Human Subjects.

### 2.2. Samples

This case-control study was conducted with samples from 233 unrelated patients with NSCLO and 568 with NSCLP, totaling 801 patients with NSCL±P, and 881 healthy control individuals. Patients with NSCL±P were carefully examined and screened for the presence of associated anomalies or syndromes by the specialized team of the four associated centers for rehabilitation of craniofacial anomalies in the Brazilian Associação de Portadores de Fissura Lábio Palatal (APOFILAB, Cascavel-PR), the Centro de Atendimento Integral ao Fissurado Labiopalatal (CAIF, Curitiba-PR), both located in the South of Brazil, the Centro Pró-Sorriso (Hospital Alzira Velano, UNIFENAS, Alfenas-MG), located in Southeastern Brazil, and the Centro de Reabilitação de Anomalias Crânio Faciais (Hospital Santo Antônio, Salvador-BA), located in Northeastern Brazil. The control group consisted of samples from the same geographic areas and it was composed of healthy individuals with no physical illness, psychiatric, birth defects or a family history of orofacial clefts.

### 2.3. Genotyping and Assessment of Genomic Ancestry

Genomic DNA was extracted from oral mucosa cells obtained by mouthwash with a 3% sucrose solution or by scraping the oral mucosa using a salting-out protocol [17]. PCR-based genotyping of rs1046117 (C_3269911_20), rs3769825 (C_1226568_10), and rs243836 (C_3225973_10) were performed on the StepOnePlus Real-Time PCR platform (Applied Biosystems, Foster City, CA, USA) using TaqMan 5′-exonuclease allelic discrimination assays (Assay-on-Demand service, Applied Biosystems). For genotyping validation, 10% of the sample was randomly selected and the reactions were repeated, revealing a reproducibility of 100%. The genotyping of rs1546124, rs8061351, rs2326398 and rs4783099 in *CRISPLD2* was previously reported [13].

Each sample was independently genotyped for 40 biallelic short insertion-deletion polymorphisms (INDELs), which were previously validated as ancestry informative markers of the Brazilian population [13]. The genomic ancestry of each individual in the case-control study was determined with the Structure software version 2.3.4 [18], applying the model of K = 3 for parental populations based on the tri-hybrid origin of the Brazilian population.

### 2.4. Statistical Analysis

The differences between groups were analyzed by chi-square test (for sex) or Kruskal-Wallis test (for genomic ancestry proportions). Genotype distributions were assessed for the Hardy-Weinberg equilibrium (HWE) in the control group by the chi-square test. The multiple logistic regression analysis under unrestricted, dominant and recessive genetic models was performed with the SNPassoc package in the R software, considering sex and ancestry proportions as potential confounders. For those analyses, a Bonferroni-adjusted *p* value threshold of ≤0.01 was applied.

Pairwise SNP-SNP interactions among *FOS*, *CASP8*, *MMP2* and *CRISPLD2* (rs1546124, rs8061351, rs2326398 and rs4783099) were performed using the model-based multifactor dimensionality reduction (mbmdr) test (mbmdr package for R) adjusted by sex and genomic ancestry and applying cross-validation and 1000 permutation strategies to reject false-positive interactions [19]. STRING, the protein-protein interaction network software, was applied to investigate the functional interactions among the examined genes (https://string-db.org/, accessed on 13 December 2022).

## 3. Results

Individual variations in the genetic ancestry proportions were detected, but the groups were not statistically different (Figure 1). All groups showed a higher prevalence of European ancestry compared to African and Amerindian. Regarding sex, the frequency of males was significantly higher in NSCL±P (n = 451, 56.3%, *p* < 0.0005) and in NSCLP (n = 326, 57.4%, *p* < 0.0005) than in the control group (n = 421, 47.8%). No significant difference between the control and NSCLO (n = 125, 53.6%) was observed. The genotype call rate ranged from 96.7% to 99.7%, and the genotype distributions had no derivation from Hardy-Weinberg equilibrium in the control group for all SNPs (Table 1).

The *FOS* rs1046117 polymorphism revealed significant associations (Table 2). The frequency of the C allele was significantly higher in patients with NSCLO than in individuals of the control group (23.5% vs. 19.6%, *p* = 0.004), with the OR in heterozygotes of 1.59 (95% CI: 1.16–2.18, *p* = 0.003) and dominant model of 1.50 (95% CI: 1.10–2.02, *p* = 0.007). The variant C allele was also more frequent in NSCL±P than in controls (22.5% vs. 19.6%), yielding an OR of 1.19 (95% CI: 1.00–1.41) and a nominal *p* value of 0.04, which did not resist to correction for multiple comparison tests. This same trend was observed for the TC heterozygotes, with an OR of 1.25 (95% CI: 1.01–1.55, *p* = 0.04), and the dominant genetic model, with an OR of 1.25 (95% CI: 1.02–1.53, *p* = 0.03), that were more frequent in NSCL±P compared to controls, but the significance did not remain after application of Bonferroni correction for multiple tests. There was no evidence of allelic or genotypic associations of *CASP8* rs3769825 (Table 3) and *MMP2* rs243836 (Table 4) with the susceptibility to NSCL±P, NSCLO and NSCLP. The stratified analysis of the samples by genomic ancestry (patients with high European ancestry and patients with high African ancestry) showed no significant results at a Bonferroni threshold (Appendix A).

As the SNPs rs1546124, rs8061351, rs2326398 and rs4783099 in *CRISPLD2* were previously analyzed in this same case-control sample [13], we sought to verify whether SNP-SNP interactions among variants in *CRISPLD2* and genes of its pathway could increase the prediction risk for nonsyndromic orofacial clefts. All possible combinations of pairs were analyzed, but only those with nominal *p* values are depicted in Table 5 (*p* < 0.05). For NSCL±P, the interactions containing *CASP8* rs3769825 and *FOS* rs104617 (p_perm_ = 0.01) and *CASP8* rs3769825 and *CRISPLD2* rs8061351 p_perm_ = 0.02) were found to be significant after permutation tests. The *FOS* rs1046117 interactions with *MMP2* rs243836 (p_perm_ = 0.03) and with *CASP8* rs3769825 (p_perm_ = 0.05) potentially increased the risk of NSCLO, whereas the interactions between rs8061351 in *CRISPLD2* with *CASP8* rs3769825 (p_perm_ = 0.02) or with *MMP2* rs243836 (p_perm_ = 0.04) increased the risk for NSCLP.

## 4. Discussion

Evidence has suggested that *CRISPLD2,* and genes in its pathway, may play important roles during craniofacial development, and the polymorphic variants in these genes may influence their function, contributing to nonsyndromic orofacial cleft susceptibility [7,8,9,10,11,12,13]. We have demonstrated, in a previous study, the association of *CRISPLD2* variants and NSCL±P risk, with a clear influence on the individual genomic ancestry [13]. This influence was observed in other studies, with *CRISPLD2* representing a candidate gene for Caucasian, Hispanic, African and Chinese populations [7,8,9,20,21], but not for individuals of Italian or Indian ancestry [22,23]. The current study explored whether polymorphic variants in *CRISPLD2*-pathway genes such as *FOS*, *CASP8* and *MMP2* individually or interacting with *CRISPLD2* contribute to NSCL±P susceptibility in the Brazilian population. Although in *CASP8* and *MMP2,* SNPs were not associated with the risk of NSCL±P, a clear tendency between the risk allele of *FOS* rs1046117 and NSCL±P was observed. Further stratified analysis revealed that the *FOS* rs1046117 C allele, TC heterozygous genotype and TC/CC genotype, representing the dominant model, significantly increased the risk of NSCLO, but not of NSCLP.

Chiquet et al. demonstrated that Fos is abundantly expressed in the orofacial region during zebrafish development, and the *FOS* rs1046117 C allele is significantly associated with an increased risk of NSCL±P in non-hispanic white families [11]. In a dimeric form with JUN, ATF or MAF, FOS forms the AP-1 transcription complex, which is implicated in the control of proliferation, differentiation, apoptotic cell death and many other important events associated with both normal development and tumorigenesis [24,25]. The rs1046117, characterized by a T to C transition at nucleotide position 252, represents a synonymous genetic variant with no alteration on the amino-acid sequence of the protein (Reference SNP Report, https://www.ncbi.nlm.nih.gov/snp/?term=rs1046117, accessed on 13 December 2022). However, the predicting effect of rs104117 on protein function was verified in the sorting intolerant from tolerant (SIFT) [26] and combined annotation-dependent depletion (CADD) [27], and both software revealed damaging scores (0.122 for SIFT and 0.867 for CADD) for the protein function in the presence of the variant allele. Consequently, a covered impact of polymorphism on the gene function, including translation efficiency and RNA stability, or affecting the AP-1 complex structure and subsequently its activity, is possible. It still is possible that this SNP belongs to a region within the gene that acts as a cis-regulatory element regulating the transcription of neighboring genes. On the other hand, as rs1046117 belongs to a large linkage disequilibrium block, its association detected in this study may potentially rely on a causative variant in this block.

In the past, genetic studies in nonsyndromic orofacial clefts have mostly focused on the analysis of individual SNPs. Although this approach has allowed the discovery of important candidates for nonsyndromic orofacial cleft risk, it clearly does not uncover potential interactions among them. SNP-SNP interaction analysis applied in this study, representing epistasis, revealed important interactions that predicted the risk of nonsyndromic orofacial clefts. The significant interactions containing *FOS* rs1046117 with *CASP8* rs3769825 in both NSCL±P and NSCLO or with *MMP2* rs243836 in NSCLO classified correctly high-risk and low-risk genotypes. Although MMP2 is expressed during craniofacial development, including normal palate fusion process, and MMP2 knockout mice display many craniofacial defects due to dysregulated osteoblast and osteoclast differentiation [28], the specific expression of MMP2 during lip development has never been described. However, due to its importance to extracellular matrix remodeling, facilitating angiogenesis and cellular migration [12,29,30], the expression of MMP2 during lip development is expected. MMP2 expression levels are known to be regulated by the AP-1 transcription factor, which is composed by association of Fos and Jun family members, and a large variety of cytokines and growth factors can trigger cell signaling culminating in MMP2 promoter activation by the convergence at the AP-1 [31]. Although evidence from both animal and human studies support a role for MMP2 as a candidate gene in the occurrence of nonsyndromic orofacial clefts, only one study has explored its genetic variants in NSCL±P risk [32]. During lip and palate morphogenesis, processes such as apoptosis play important roles in different periods, and some craniofacial abnormalities have been attributed to irregular activation of the apoptotic cascade [33,34]. After activation, caspase 8 activates effector caspases, inducing the apoptotic caspase cascade which is essential to medial edge epithelium degradation and subsequent palatal fusion [35]. Collectively, the results suggested that *FOS* rs1046117 may affect NSCL±P and NSCLO susceptibility by interacting with *MMP2* and *CASP8*.

For NSCLP, interactions with significant *p* values after correction with 1000 permutations involved *CRISPLD2* rs8061351 with *CASP8* rs3769825 and with *MMP2* rs243836, but not with *FOS*. CRISPLD2 overexpression promoted apoptosis of lung fibroblasts after activation of multiple proapoptotic genes and caspase activities, and also regulated migration and extracellular matrix genes that modulate lung development and repair, including MMP [36]. The loss-of-function strategy using morpholino targeting *Crispld2* revealed a direct control of both *Casp8 and Mmp2*, supporting a regulatory effect of CRISPLD2 in events dependent on interactions between CRISPLD2-CASP8 and CRISPLD2-MMP2 [11]. Together, our findings show that variants in the CRISPLD2 pathway may influence the risk of NSCLP through potential epistatic interaction.

The study has strengths and limitations. Among the strengths we can highlight its multicenter design, enrolling samples from distinct regions of Brazil, which brings a better representation of the Brazilian population, and the use of robust statistical approaches with control for confounding effects including sex and ancestry proportions and application of correction for multiple comparison tests such as Bonferroni threshold and 1000 permutation, which reduce spurious results. The limitations include the test of only one SNP in each of the candidate genes, the lack of characterization of the impact of SNPs on function of the encoded proteins, the limited power in the stratification analyses due to modest sample size, though the effect of *FOS* SNP was highlighted when NSCLO was separated from NSCLP, and the absence of environmental factors, which could exert important roles under gene-environment interactions.

## 5. Conclusions

Our results demonstrated the association of the genetic variant rs1046117 in *FOS* with NSCLO in the Brazilian population. Based on the collective information and biological plausibility (Figure 2), genes in the CRISPLD2 pathway are likely to be involved in the occurrence of NSCL±P and they must be studied further in large and independent datasets, providing more valid results for clinical decision-making.

## Figures and Tables

**Figure 1 dentistry-11-00007-f001:**
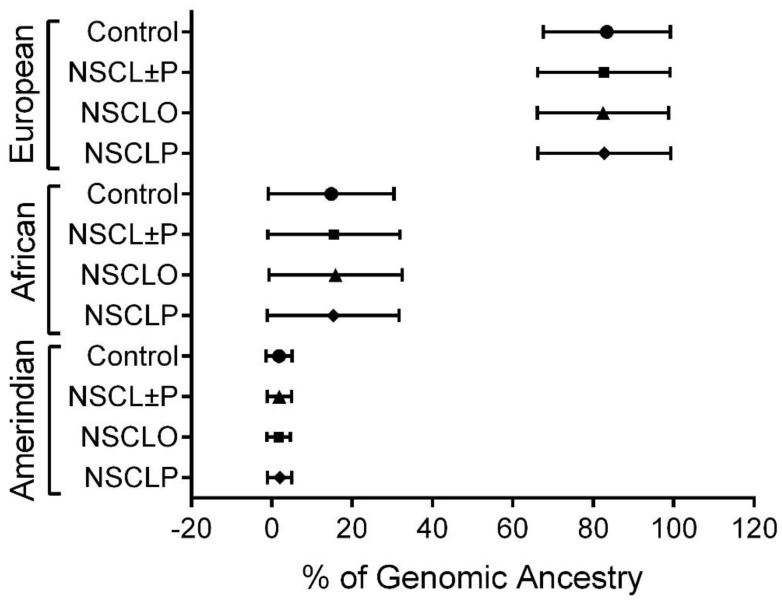
Genomic ancestry proportions from the groups of patients with nonsyndromic cleft lip with or without cleft palate (NSCL±P), nonsyndromic cleft lip only (NSCLO) and nonsyndromic cleft lip and palate (NSCLP) and control. Circle was designed to control group, square to NSCL±P, triangle to NSCLO and diamond to NSCLP.

**Figure 2 dentistry-11-00007-f002:**
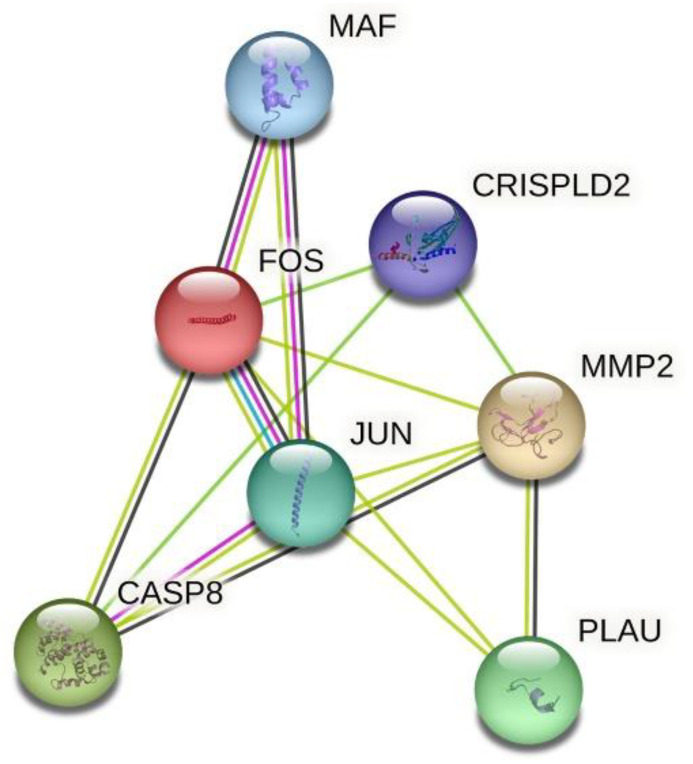
Protein-protein interaction network of the examined genes. A clear node involving FOS, CASP8, MMP2 and CRISPLD2 was generated using STRING. Different colors represent different levels of evidence of connection between proteins. Light blue represents curated databases, purple experimental evidence, light green evidence from text mining and black co-expression. This analysis had an average confidence score of 0.719, suggesting a low rate of false-positive interactions.

**Table 1 dentistry-11-00007-t001:** Characteristics of single-nucleotide polymorphisms (SNPs), frequency of the minor alleles, Hardy-Weinberg equilibrium and genotyping call rates.

Gene	SNP	Position	Allele *	MAF	HWE (*p* Value)	Call Rate
*FOS* (chromosome 14)	rs1046117	75279987	T/**C**	0.196	0.06	96.7%
*CASP8* (chromosome 2)	rs3769825	201246657	A/**G**	0.460	0.46	98.3%
*MMP2* (chromosome 16)	rs243836	55500324	G/**A**	0.464	0.74	98.8%
*CRISPLD2* (chromosome 16)	rs1546124	84838445	A/**G**	0.322	0.57	99.6%
	rs8061351	84849496	T/**C**	0.408	0.34	99.7%
	rs2326398	84869111	A/**G**	0.377	0.54	99.7%
	rs4783099	84907723	C/**T**	0.365	0.07	99.7%

* Minor allele in bold. MAF: Minor allele frequency. HWE: Hardy-Weinberg equilibrium.

**Table 2 dentistry-11-00007-t002:** Association between *FOS* rs1046117 and the risk of nonsyndromic cleft lip with or without cleft palate (NSCL±P), nonsyndromic cleft lip only (NSCLO) and nonsyndromic cleft lip and palate (NSCLP). *p* values were adjusted for covariates by logistic regression analysis.

	Control	NSCL±P	OR (95% CI)/ *p* Value	NSCLO	OR (95% CI)/ *p* Value	NSCLP	OR (95% CI)/ *p* Value
Allele							
T	80.4%	78.0%	Reference	76.5%	Reference	79.5%	Reference
C	19.6%	22.0%	1.19 (1.00–1.41)/0.04	23.5%	1.28 (1.10–1.64)/0.004	20.5%	1.15 (0.95–1.39)/0.13
Genotype							
TT	65.9%	60.6%	Reference	56.4%	Reference	62.4%	Reference
TC	29.0%	33.6%	1.25 (1.01–1.55)/0.04	39.5%	1.59 (1.16–2.18)/0.003	31.2%	1.13 (0.89–1.44)/0.30
CC	5.1%	5.8%	1.22 (0.79–1.90)/0.36	4.1%	0.95 (0.45–1.99)/0.90	6.4%	1.35 (0.84–2.16)/0.21
Dominant (TT/TC + CC)	65.9%/34.1%	60.6%/39.4%	1.25 (1.02–1.53)/0.03	56.4%/43.6%	1.50 (1.10–2.02)/0.007	62.4%/37.6%	1.16 (0.93–1.46)/0.18
Recessive (TT + TC/CC)	94.9%/5.1%	94.2%/5.8%	1.14 (0.74–1.75)/0.55	95.9%/4.1%	0.80 (0.39–1.67)/0.54	93.6%/6.4%	1.30 (0.82–2.06)/0.27

**Table 3 dentistry-11-00007-t003:** Association between *CASP8* rs3769825 and the risk of nonsyndromic cleft lip with or without cleft palate (NSCL±P), nonsyndromic cleft lip only (NSCLO) and nonsyndromic cleft lip and palate (NSCLP). *p* values were adjusted for covariates by logistic regression analysis.

	Control	NSCL±P	OR (95% CI)/ *p* Value	NSCLO	OR (95% CI)/ *p* Value	NSCLP	OR (95% CI)/ *p* Value
Allele							
A	54.0%	54.8%	Reference	54.9%	Reference	54.8%	Reference
G	46.0%	45.2%	0.89 (0.77–1.02)/0.09	45.1%	0.83 (0.67–1.02)/0.08	45.2%	0.91 (0.78–1.06)/0.24
Genotype							
AA	28.4%	31.5%	Reference	34.7%	Reference	30.2%	Reference
AG	51.0%	50.6%	0.91 (0.73–1.14)/0.38	47.6%	0.76 (0.54–1.05)/0.09	51.9%	0.98 (0.77–1.26)/0.86
GG	20.6%	17.9%	0.80 (0.60–1.06)/0.12	17.7%	0.71 (0.46–1.09)/0.11	17.9%	0.84 (0.61–1.15)/0.27
Dominant (AA/AG + GG)	28.4%/71.6%	31.5%/68.5%	0.88 (0.71–1.09)/0.22	34.7%/65.3%	0.74 (0.54–1.02)/0.06	30.2%/69.8%	0.94 (0.74–1.19)/0.61
Recessive (AA + AG/GG)	79.4%/20.6%	82.1%/17.9%	0.85 (0.66–1.09)/0.19	82.3%/17.7%	0.85 (0.58–1.24)/0.38	82.1%/17.9%	0.85 (0.65–1.12)/0.23

**Table 4 dentistry-11-00007-t004:** Association between *MMP2* rs243836 and the risk of nonsyndromic cleft lip with or without cleft palate (NSCL±P), nonsyndromic cleft lip only (NSCLO) and nonsyndromic cleft lip and palate (NSCLP). *p* values were adjusted for covariates by logistic regression analysis.

	Control	NSCL±P	OR (95% CI)/ *p* Value	NSCLO	OR (95% CI)/ *p* Value	NSCLP	OR (95% CI)/ *p* Value
Allele							
G	53.6%	53.2%	Reference	53.6%	Reference	52.9%	Reference
A	46.4%	46.8%	1.04 (0.91–1.19)/0.53	46.4%	0.99 (0.80–1.22)/0.95	47.1%	1.06 (0.91–1.23)/0.40
Genotype							
GG	28.4%	28.0%	Reference	29.6%	Reference	27.4%	Reference
GA	50.3%	48.9%	0.96 (0.76–1.21)/0.76	48.3%	0.90 (0.64–1.27)/0.55	49.1%	0.98 (0.76–1.26)/0.91
AA	21.3%	23.1%	1.06 (0.81–1.40)/0.73	22.1%	0.98 (0.65–1.48)/0.91	23.5%	1.10 (0.81–1.48)/0.64
Dominant (GG/GA + AA)	28.4%/71.6%	28.0%/72.0%	0.99 (0.80–1.23)/0.93	29.6%/70.4%	0.92 (0.67–1.27)/0.62	27.4%/72.6%	1.01 (0.80–1.29)/0.91
Recessive (GG + GA/AA)	78.7%/21.3%	76.9%/23.1%	1.09 (0.86–1.38)/0.47	77.9%/22.1%	1.04 (0.73–1.48)/0.82	76.5%/23.5%	1.11 (0.86–1.43)/0.42

**Table 5 dentistry-11-00007-t005:** SNP-SNP interactions among *CRISPLD2*, *FOS*, *CASP8* and *MMP2* assessed by the model-based multifactor dimensionality reduction (*mbmdr)* test.

	SNP1	SNP2	NH ^a^	betaH ^b^	NL ^c^	betaL ^d^	*p* Value ^e^	Perm. *p* Value ^f^
NSCL±P								
	rs1046117 (*FOS*)	rs3769825 (*CASP8*)	2	0.5426	2	−0.2482	0.0005	0.01
	rs8061351 (*CRISPLD2*)	rs3769825 (*CASP8*)	2	0.5479	0	NA	0.001	0.02
	rs8061351 (*CRISPLD2*)	rs243836 (*MMP2*)	1	0.7808	1	−0.4374	0.003	0.06
	rs4783099 (*CRISPLD2*)	rs243836 (*MMP2*)	1	1.0643	0	NA	0.01	0.12
	rs1546124 (*CRISPLD2*)	rs1046117 (*FOS*)	2	0.6675	0	NA	0.01	0.11
	rs2326398 (*CRISPLD2*)	rs243836 (*MMP2*)	1	0.4507	0	NA	0.01	0.18
NSCLO								
	rs1046117 (*FOS*)	rs243836 (*MMP2*)	1	0.6455	0	NA	0.003	0.03
	rs1046117 (*FOS*)	rs3769825 (*CASP8*)	1	0.6822	1	−0.4221	0.004	0.05
	rs2326398 (*CRISPLD2*)	rs3769825 (*CASP8*)	1	1.1374	0	NA	0.003	0.06
	rs8061351 (*CRISPLD2*)	rs3769825 (*CASP8*)	1	0.8242	0	NA	0.01	0.16
	rs1546124 (*CRISPLD2*)	rs1046117 (*FOS*)	1	0.9427	0	NA	0.03	0.21
	rs8061351 (*CRISPLD2*)	rs243836 (*MMP2*)	1	0.5800	1	−0.3993	0.04	0.31
NSCLP								
	rs8061351 (*CRISPLD2*)	rs3769825 (*CASP8*)	2	0.6082	0	NA	0.0008	0.02
	rs8061351 (*CRISPLD2*)	rs243836 (*MMP2*)	2	0.6699	1	−0.4583	0.002	0.04
	rs1046117 (*FOS*)	rs3769825 (*CASP8*)	1	0.9145	0	NA	0.007	0.10
	rs2326398 (*CRISPLD2*)	rs243836 (*MMP2*)	1	0.5372	0	NA	0.008	0.13
	rs1546124 (*CRISPLD2*)	rs243836 (*MMP2*)	1	0.5256	0	NA	0.01	0.16
	rs4783099 (*CRISPLD2*)	rs243836 (*MMP2*)	1	1.0860	1	−0.3102	0.01	0.17
	rs1546124 (*CRISPLD2*)	rs1046117 (*FOS*)	1	0.9437	0	NA	0.03	0.25

^a^ Number of significant high-risk genotypes in the interaction. ^b^ Regression coefficient in step2 for high-risk exposition. ^c^ Number of significant low-risk genotypes in the interaction. ^d^ Regression coefficient in step2 for low-risk exposition. ^e^
*p* value for the interaction model adjusted for covariates. ^f^ Permutation *p* value for the interaction model.

## Data Availability

The original data that support the findings of this study are available from the corresponding author upon reasonable request.

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
