# Peer review of "Brazilian Multiethnic Association Study of Genetic Variant Interactions among FOS, CASP8, MMP2 and CRISPLD2 in the Risk of Nonsyndromic Cleft Lip with or without Cleft Palate"

_dentistry, 2022, doi:10.3390/dj11010007_

Round 1

Reviewer 1 Report

In "Brazilian multiethnic association study of genetic variant inter-2 actions among FOS, CASP8, MMP2 and CRISPLD2 in the risk of nonsyndromic cleft lip with or without cleft palate" Machado and colleagues report the results of an association study of CRISPLD2 (cysteine-rich secretory protein LCCL domain containing 2) and genes belonging to its activation pathway in a large cohort of patients (n=801) with NSCL±P, and healthy  control individuals (n=881).

While it is focused on candidate genes for orofacial clefting, not yet thoroughly explored and the study is performed in a large cohort this study is of importance.

I have a few minor comments.

- Line  96-97 sample:

“ individuals. Patients with NSCL±P were carefully examined and screened for the presence of associated anomalies or syndromes by the specialized team associated centers for rehabilitation of craniofacial anomalies in the Brazil

Differentiating non-syndromic CLA/P from syndromic CLA/P cases on clinical phenotypes, more and more challenging. Therefore, NGS based gene panel testing, targeting known genes associated with CL±P, is nowadays more common in clinical genetic counseling of CL±P. Moreover, there is increasing evidence that rare and common variants are involved in both Mendelian and complex forms of CL±P.

Is diagnostic DNA analysis performed in these patients or is this cohort screened for causative pathogenic gene-variants?

-  line 70-71

Although putative roles for those genes are expected during orofacial development, only MMP2 has been described as essential for normal palatogenesis [12]”

I don’t understand this sentence is in italic.

- line 68-71 and line 215-216 and 232-236 Introduction and Discussion

“ which was associated, among other putative candidates, with dysregulated expression of FOS (Fos proto-oncogene, which encodes a leucine zipper protein that can dimerize with  proteins of the JUN family, thereby forming the transcription factor complex AP-1), CASP8 (caspase 8) and MMP2 (matrix metalloproteinase 2) [11]”

“In a dimeric form with  JUN, ATF or MAF, FOS forms the AP-1 transcription complex, which is implicated in the control of proliferation, differentiation, apoptotic cell death and many other important events associated with both normal development and tumorigenesis”

“The MMP2 expression levels are known to be  regulated by AP-1 transcription factor [26], and during the normal palate fusion process, MMP2 expression is important to extracellular matrix remodeling, facilitating angiogenesis and cellular migration”

In above paragraphs the interaction between several genes are discussed.

A graphic depicting these relations might be illustrative and helps to interpret the results and discussion.

Line 197 - 266 Discussion

This study reveals:

For NSCLO, the FOS rs1046117 interactions with MMP2 rs243836 and FOS rs1046117 with CASP8 rs3769825 potentially increased the risk.

As stated in line 72 and 232-236 MMP2 has been described  as essential for normal palatogenesis. However interaction of FOS rs1046117 with MMP2 rs243836 increased the risk for NSCLO.

Can the authors elaborate on the increased risk for NSCLO by interactions of FOS with MMP2? Is MMP2 also expressed in lip tissue? Might time span of expression play a role?

Line 267

5. Coclusions – typo -> Conclusions

Author Response

We would like to thank the reviewer for the encouraging comments on the achievements and quality of study, and for the suggestions to improve it. The point-by-point answers are below of each of your questions/comments.

- Line 96-97 sample:

individuals. Patients with NSCL±P were carefully examined and screened for the presence of associated anomalies or syndromes by the specialized team associated centers for rehabilitation of craniofacial anomalies in the Brazil

Differentiating non-syndromic CLA/P from syndromic CLA/P cases on clinical phenotypes, more and more challenging. Therefore, NGS based gene panel testing, targeting known genes associated with CL±P, is nowadays more common in clinical genetic counseling of CL±P. Moreover, there is increasing evidence that rare and common variants are involved in both Mendelian and complex forms of CL±P.

Is diagnostic DNA analysis performed in these patients or is this cohort screened for causative pathogenic gene-variants?

Unfortunately, those patients are not subjected to any genetic test, unless the suspicion of a syndrome is raised. All cases with possible suspicion of syndromic cleft were not included in our study. In Brazil, the differentiation between syndromic and nonsyndromic orofacial clefts is based on a careful clinical examination by trained clinical geneticists.

- line 70-71

Although putative roles for those genes are expected during orofacial development, only MMP2 has been described as essential for normal palatogenesis [12]”

I don’t understand this sentence is in italic.

We are sorry for our mistake. There was no specific reason, and it was corrected.

- line 68-71 and line 215-216 and 232-236 Introduction and Discussion

“ which was associated, among other putative candidates, with dysregulated expression of FOS (Fos proto-oncogene, which encodes a leucine zipper protein that can dimerize with  proteins of the JUN family, thereby forming the transcription factor complex AP-1), CASP8 (caspase 8) and MMP2 (matrix metalloproteinase 2) [11]”

“In a dimeric form with  JUN, ATF or MAF, FOS forms the AP-1 transcription complex, which is implicated in the control of proliferation, differentiation, apoptotic cell death and many other important events associated with both normal development and tumorigenesis”

“The MMP2 expression levels are known to be  regulated by AP-1 transcription factor [26], and during the normal palate fusion process, MMP2 expression is important to extracellular matrix remodeling, facilitating angiogenesis and cellular migration”

In above paragraphs the interaction between several genes are discussed.

A graphic depicting these relations might be illustrative and helps to interpret the results and discussion.

Many thanks for this very good suggestion. We verified the protein-protein interactions using the STRING software (protein-protein interaction networks, https://string-db.org/). The interactions for the examined genes (FOS, CASP8, MMP2 and CRISPLD2) are depicted in Figure 2, which was included in the revised manuscript. A short description of analysis was also included in Materials and Methods.

- Line 197 - 266 Discussion

This study reveals:

For NSCLO, the FOS rs1046117 interactions with MMP2 rs243836 and FOS rs1046117 with CASP8 rs3769825 potentially increased the risk.

As stated in line 72 and 232-236 MMP2 has been described  as essential for normal palatogenesis. However interaction of FOS rs1046117 with MMP2 rs243836 increased the risk for NSCLO.

Can the authors elaborate on the increased risk for NSCLO by interactions of FOS with MMP2? Is MMP2 also expressed in lip tissue? Might time span of expression play a role?

These are very astute questions. Although MMP2 is expressed during craniofacial development and MMP2 knockout mice display many craniofacial defects due to both dysregulated osteoblast and osteoclast differentiation, the specific expression of MMP-2 during lip development has never been described. However, due to its importance in tissue remodeling, the expression is expected. As mentioned, lines 234-237, MMP-2 gene is responsive to AP-1 transcription factor, which is composed by association of Fos and Jun family members, and a large variety of cytokines and growth factors, which are active during development, can trigger cell signaling culminating in transactivation of MMP-2 promoter by the convergence at the AP-1. In the revised manuscript, we have further discussed the connection between FOS and MMP2 in the risk of NSCLO, as suggested. Nevertheless, we limited this discussion in the interest of focus (avoiding too much speculation), because we did not perform functional analyses to confirm this relationship. The following sentence was included:

“Although MMP2 is expressed during craniofacial development, including normal palate fusion process, and MMP2 knockout mice display many craniofacial defects due to dysregulated osteoblast and osteoclast differentiation [28], the specific expression of MMP2 during lip development has never been described. However, due to its importance to extracellular matrix remodeling, facilitating angiogenesis and cellular migration [12,29,30], the expression of MMP2 during lip development is expected. MMP2 expression levels are known to be regulated by AP-1 transcription factor, which is composed by association of Fos and Jun family members, and a large variety of cytokines and growth factors can trigger cell signaling culminating in MMP2 promoter activation by the convergence at the AP-1 [31]. Although evidence from both animal and human studies support a role for MMP2 as a candidate gene in the occurrence of nonsyndromic orofacial clefts, only one study has explored its genetic variants in NSCL±P risk [32].”

- Line 267

5. Coclusions – typo -> Conclusions

Thank you for highlighting this typo. It was corrected.

Reviewer 2 Report

The study was very well designed, written and organized, however few points have to be amended in the paper

1) Introduction

Very well written but it needs further amendments

1) The sentence from 51-53 (1 out 700 is affected) need a reference.

2)  line 71-73 is written in italic, please correct it.

2) Materials and methods

1) Samples paragraph, please add the age range and the gender of all the participated subjects.

2) Please add a supplementary table showing the primer sequence used and the product size.

3)Results

Very well written and interpreted.

4) Discussion

1) Line 218-226, explain a lot of information that comes from previous studies without any reference, beside you have mentioned that rs104117 is synonymous SNP that did not alter the protein, In your explanation you have to add to which extent this mutation is pathogenic using CADD or SIFT.

2) The sample size and sequencing one SNP in each gene should be added to the limitation of the study.

Author Response

We would like to thank the reviewer for its positive evaluation of study, and to provide us with timely and important suggestions to improve our manuscript. The point-by-point answers are below of each of your questions/comments.

1) Introduction

Very well written but it needs further amendments

1) The sentence from 51-53 (1 out 700 is affected) need a reference.

As requested, a reference was included to support our statement about worldwide prevalence of nonsyndromic orofacial clefts.

2)  line 71-73 is written in italic, please correct it.

Sorry for our mistake. This was corrected.

2) Materials and methods

1) Samples paragraph, please add the age range and the gender of all the participated subjects.

We have described in the first paragraph of the results, the sex of patients, highlighting the significant differences (now on lines 142-145). As a congenital disease, age is not a relevant parameter in studies involving genetic features in orofacial clefts. All samples were collected from children and teenagers.

2) Please add a supplementary table showing the primer sequence used and the product size.

It is not clear the specific question of the reviewer. Regarding SNP genotyping, we have applied the TaqMan technology as assays-on-demand from Applied Biosystems, and we do not have access to the sequences of the primers and probes. Details such as the SNP ID (for example, C_3269911_20 for rs1046117) were already included. The sequences of the primers and PCR product sizes used for the estimation of the genomic ancestry are described in details on Supplementary Tables 1 and 2 of study performed by Bastos-Rodrigues et al. (2006). This information is included in the reference 13 of the manuscript, as well as the methodologic strategy. With all respect, we did not repeat these information in the interest of focus and space.

Bastos-Rodrigues L, Pimenta JR, Pena SDJ. The Genetic Structure of Human Populations Studied Through Short Insertion-Deletion Polymorphisms. Ann Hum Genet. 2006 Sep;70(5):658-665. doi: 10.1111/j.1469-1809.2006.00287.x.

3) Results

Very well written and interpreted.

Many thanks for your positive comment.

4) Discussion

1) Line 218-226, explain a lot of information that comes from previous studies without any reference, beside you have mentioned that rs104117 is synonymous SNP that did not alter the protein, in your explanation you have to add to which extent this mutation is pathogenic using CADD or SIFT.

Regarding the first part of your comment, most of the information was collected from the Reference SNP Report (RefSNP), belonging to The National Center for Biotechnology Information (NCBI-NIH). This information was included in the revised manuscript.

As suggested, we have verified the effect of the synonymous variation on the predicting functional effects of the protein, and both algorithms showed scores of moderate pathogenicity. The following paragraph was included in the revised manuscript:

“The predicting effect of rs104117 on protein function was verified in the sorting intolerant from tolerant (SIFT) [26] and combined annotation dependent depletion (CADD) [27], and both software revealed damaging scores (0.122 for SIFT and 0.867 for CADD) for the protein function in the presence of the variant allele.”

2) The sample size and sequencing one SNP in each gene should be added to the limitation of the study.

As suggested, we have highlighted those features as study limitations.